# Using Sentinel-1, Sentinel-2, and Planet satellite data to map field-level tillage practices in smallholder systems

Yin Liu[1], Preeti Rao[1,2], Weiqi Zhou[1], Balwinder Singh [3,4], Amit K. Srivastava[5], Shishpal P. Poonia[3], Derek Van Berkel[1], Meha Jain[1] *

1 School for Environment and Sustainability, University of Michigan, Ann Arbor, MI, United States of America, 2 Center for Climate Change and Sustainability, Azim Premji University, Bengaluru, India, 3 International Mazie and Wheat Improvement Center (CIMMYT)-India Officer, New Delhi, India, 4 Department of Primary Industries and Regional Development, Northam, Western Australia, Australia, 5 IRRI South Asia Regional Centre (ISARC), NSRTC Campus, Varanasi, India

* mehajain@umich.edu

**Data Availability Statement:** We have now included all minimal data sets in the Zenodo data

## Abstract

Remote sensing can be used to map tillage practices at large spatial and temporal scales. However, detecting such management practices in smallholder systems is challenging given that the size of fields is smaller than historical readily-available satellite imagery. In this study we used newer, higher-resolution satellite data from Sentinel-1, Sentinel-2, and Planet to map tillage practices in the Eastern Indo-Gangetic Plains in India. We specifically tested the classification performance of single sensor and multiple sensor random forest models, and the impact of spatial, temporal, or spectral resolution on classification accuracy. We found that when considering a single sensor, the model that used Planet imagery (3 m) had the highest classification accuracy (86.55%) while the model that used Sentinel-1 data (10 m) had the lowest classification accuracy (62.28%). When considering sensor combinations, the model that used data from all three sensors achieved the highest classification accuracy (87.71%), though this model was not statistically different from the Planet only model when considering 95% confidence intervals from bootstrap analyses. We also found that high levels of accuracy could be achieved by only using imagery from the sowing period. Considering the impact of spatial, temporal, and spectral resolution on classification accuracy, we found that improved spatial resolution from Planet contributed the most to improved classification accuracy. Overall, it is possible to use readily-available, high spatial resolution satellite data to map tillage practices of smallholder farms, even in heterogeneous systems with small field sizes.

## Introduction

Conventional tillage (CT) is typically used to prepare agricultural fields for planting as it controls weeds, removes most plant residue from the previous crop, and creates a fine seedbed [1]. Yet, CT negatively impacts long-term soil health due to increased erosion, the loss of organic

repository (https://doi.org/10.5281/zenodo.6703973).

**Funding:** This study was supported by the National Aeronautics and Space Administration Land Cover and Land Use Change Program through a grant awarded to MJ (Grant Number: NNX17AH97G).

**Competing interests:** The authors have declared that no competing interests exist.

matter, and reduced water retention [2]. Zero tillage (ZT), where seeds are planted directly into untilled fields that typically contain previous crop residue, has been shown to provide agronomic and economic benefits, including minimizing soil disturbance, reducing soil erosion, and maintaining soil cover [3–5]. Globally, the amount of land area under ZT has increased since the 1990s [6], yet quantifying the exact area under ZT has been challenging given that typical methods used to collect such information, such as censuses, are not implemented in all regions of the world due to financial, accessibility, and labor constraints [7]. This is particularly true in smallholder systems, where small-scale studies have suggested that ZT adoption rates have increased steadily in recent years [8]. Understanding ZT adoption in smallholder systems is critically important given that it has been shown to be an important way to sustainably intensify cereal grains in these systems [9–11]. Remote sensing can offer an alternative and low-cost way to quantify ZT adoption at large spatial and temporal scales.

Numerous detection techniques have been developed to map tillage practices using remote sensing [12, 13], yet these approaches may not be suitable for detecting tillage practices in smallholder systems [14, 15]. This is largely because the size of smallholder fields (< 2 ha) is typically smaller than the spatial resolution of historically-available satellite data that were used in previous studies, such as Landsat (30 m) [14, 16, 17]. Over the last five years, new higher-spatial resolution satellites, such as Sentinel-1 (10 m), Sentinel-2 (10 m), and Planet (3 m), have become available, and studies have shown that these sensors are better able to capture field-level variation of smallholder farms [18–21]. It is possible that these higher resolution sensors may be able to effectively map tillage practices in smallholder systems, yet to our knowledge, no studies exist that have used these higher resolution satellite datasets to map tillage practices in smallholder systems.

Previous studies that have used satellite data to map tillage practices have found that using multiple sensors in classification algorithms can improve accuracy. For example, Azzari et al. (2019) found that combining optical Landsat satellite data and radar Sentinel-1 data led to higher classification accuracies of tillage practices across the United States Midwest, although the contribution of Sentinel-1 data was small. In smallholder systems, it is possible that combining Planet, Sentinel-1, and Sentinel-2 satellite data may improve classification accuracies. Sentinel-1 has the advantage of being less sensitive to water vapor and cloud cover as radar data have the ability to penetrate cloud cover, providing data more regularly through time than optical sensors [22]. This is important in smallholder systems given that they are largely found throughout the tropics with periods of high rainfall and cloud cover [18]. Sentinel-2 imagery has multiple spectral bands that cover the visible near-infrared (VNIR), shortwave-infrared (SWIR), and red-edge wavelengths; red-edge spectral bands are particularly critical for mapping vegetation characteristics [23, 24] and have been found to increase the accuracy of mapping crop residue cover [25, 26]. Planet imagery has higher spatial resolution (3 m) than Sentinel-1 and Sentinel-2 imagery (10 m), which may reduce the effect of mixed pixels at field edges [21, 27].

Though previous studies have shown that classification accuracy of mapping agricultural characteristics, including crop type and yield, can be improved by using images throughout the growing season [19, 28–30], it is possible that only using images during the early part of the growing season may result in high classification accuracies for mapping tillage practices. This is because most spectral and phenological differences are likely to occur at the start of the growing season as fields are prepared and seedlings germinate. Producing accurate maps of tillage practices using only early season imagery could allow for within-season mapping of ZT areas; such information could be important for policy makers and decision-makers who could use such maps in real time [31].

This study examines the ability of Planet, Sentinel-1, and Sentinel-2 imagery to map field-level ZT and CT in smallholder farming systems. We focus our study in the state of Bihar in the Eastern Indo-Gangetic Plains in India, which is a region where ZT adoption has increased over the last decade [32], where field sizes are very small ($< 0.3$ ha on average), and where farm management practices are heterogeneous [33]. We aim to answer the following questions in this study:

1. How effectively can single sensor and multiple sensor combinations of Planet, Sentinel-1, and Sentinel-2 map field-level tillage practices of smallholder farms? Which features are the most important for classification accuracy?

2. Can we use only early season imagery to effectively map tillage practices, which can be used to provide within-season maps of ZT adoption?

3. Does improved spatial, temporal, or spectral resolution result in higher classification accuracy?

This study is one of the first to examine how well tillage practices can be mapped in smallholder systems using newer high-resolution satellite imagery. While the analyses presented in this paper are specific to our study area in India, it is likely that the broad findings from our study can be used to inform the most effective ways to map tillage practices in other smallholder systems across the globe.

## Materials

### Study area and ground truth data

The study was conducted in Arrah district, Bihar, India (25.47˚N, 84.52˚E) in 2017–2018 (Fig 1). Bihar has fertile soil and a large amount of rainfall, but its agricultural productivity is one of the lowest among all Indian states [10]. Although the adoption of ZT technology in Bihar has increased through time, it is still limited because many farmers lack awareness of ZT and do not have access to the machinery required for ZT [10]. While it is difficult to estimate the amount of area under ZT due to a lack of available ground data, surveys suggest that ZT adoption is variable across villages and can range from 0% to 98% of village land area in Bihar [10]. We focused on a 30 by 70 km$^2$ region where there was variation in tilling practices, and we collected ground truth data from 20 villages distributed across the study area. This region is predominantly comprised of smallholder fields ($< 0.3$ ha on average), with farms covering over 80% of land area and with over 80% of the region's population taking part in farming [34]. There are two main cropping seasons in this region, the monsoon (*kharif*) season, which spans from June to October and is when most farmers plant rice, and the winter (*rabi*) season, which spans from November to April and is when most farmers plant wheat [19]. Our study only focused on wheat fields planted during the winter cropping season.

Field management across this region is extremely heterogeneous, making it a complex system in which to map tillage practices using one universal algorithm. The sowing dates of wheat vary widely across the study region, ranging from mid-November to early January [35]. Wheat variety planted and input use, including fertilizer and irrigation, is also highly variable, with significant heterogeneity even across neighboring fields [19, 35]. Most farmers clear rice residues prior to sowing wheat, though some farmers leave partial residues in field. Rice residues are typically removed in this region as farmers harvest rice manually, which leaves behind little to no residue, farmers use rice residues for crop feed, and farmers use older generation ZT machinery that is less effective at planting wheat seeds within standing rice residues. Given that most CT and ZT fields are cleared of rice residues prior to wheat planting, this makes it additionally challenging to map ZT in this region.

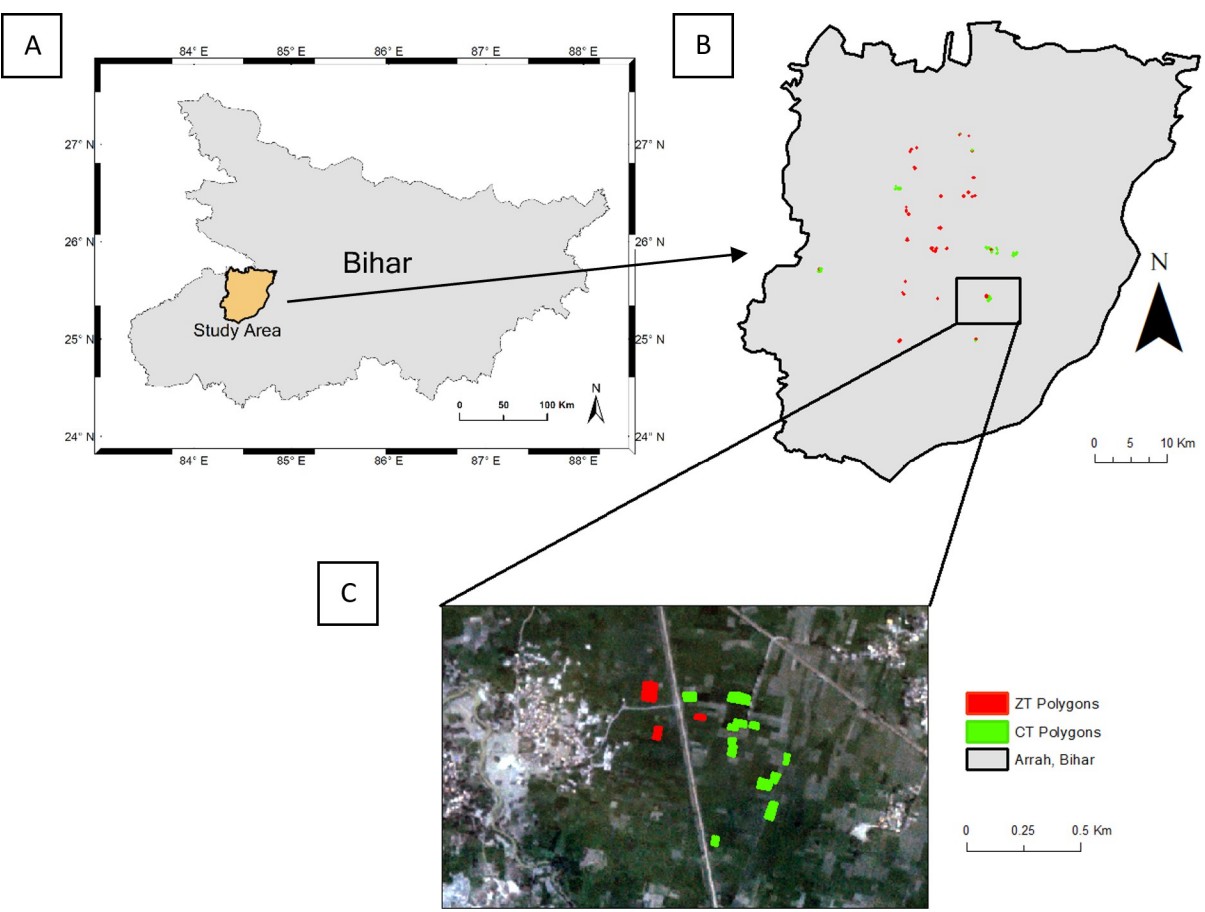

**Fig 1. Schematic diagram of the study area.** Maps showing (A) the location of our study area in Bihar, (B) our study polygons plotted in Arrah district, Bihar and (C) a zoom in of some of our polygons overlaid on Planet imagery.

Ground truth data were collected during April to May 2018 by field staff from the Cereal Systems Initiative of South Asia, which is a program under the International Maize and Wheat Improvement Center (CIMMYT-CSISA). The survey collected information on tillage practices, including whether the field was under CT or ZT, and other field management practices (See S1 Table in S1 File for survey). Example photos of ZT and CT fields are shown in Fig 2. In addition, we collected GPS locations at the four corners and at the center of each field which were later used to manually digitize field boundaries. We did this by overlaying all GPS points on high-resolution imagery in Google Earth Pro (https://www.google.com/earth/versions/#earth-pro) and manually drawing polygons that connected the four corner GPS locations. We then manually adjusted these polygons to match visible field boundaries that we could see in the high-resolution imagery. We selected the image date within Google Earth Pro that was closest to our time of survey to ensure that visible field boundaries in the imagery were consistent with the field boundaries that we surveyed on the ground. Our survey data were collected from a total of 160 fields, with 65 fields representing ZT and 95 fields representing CT.

## Satellite data

We selected the time period for our analysis by considering the timing of cropping cycles in this region as well as the phenologies of ZT and CT fields (Fig 3). Considering the timing of

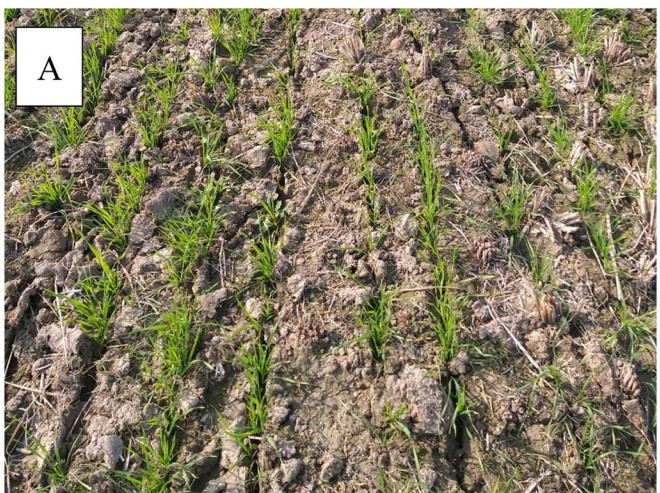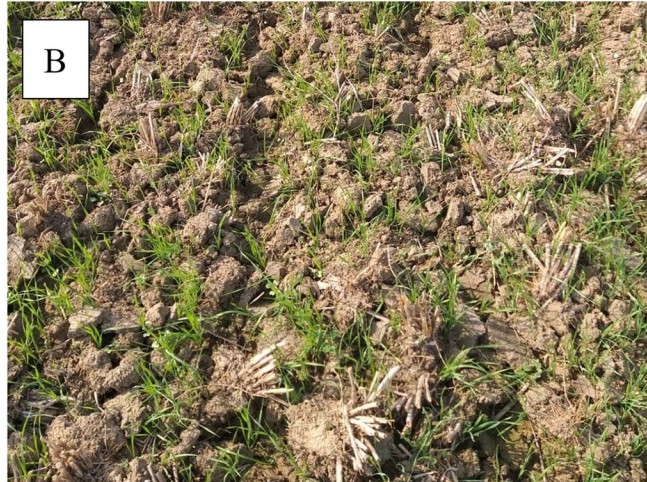

**Fig 2.** Photos of example zero tillage (A) and conventional tillage (B) fields.

cropping cycles, wheat was planted in our study area from November 11 to December 31, with tillage occurring from mid-October to early November. To ensure that we captured the full range of phenological change when the field was fallow prior to planting wheat, we used October 1st as the first date of our study period. Considering NDVI (Normalized Difference Vegetation Index) phenologies, we observed that NDVI values became similar in both ZT and CT fields after mid-February (Fig 3). Thus, we used March 1st, 2018 as the last date of our study period. We defined the sowing season as October 1 to December 31, since this time period spanned the full range of sowing dates in our dataset, and defined the full season as October 1 to March 1. From October to December, NDVI is higher in ZT fields compared to CT fields (Fig 3), likely because farmers maintain monsoon rice crop residue on ZT fields but not CT

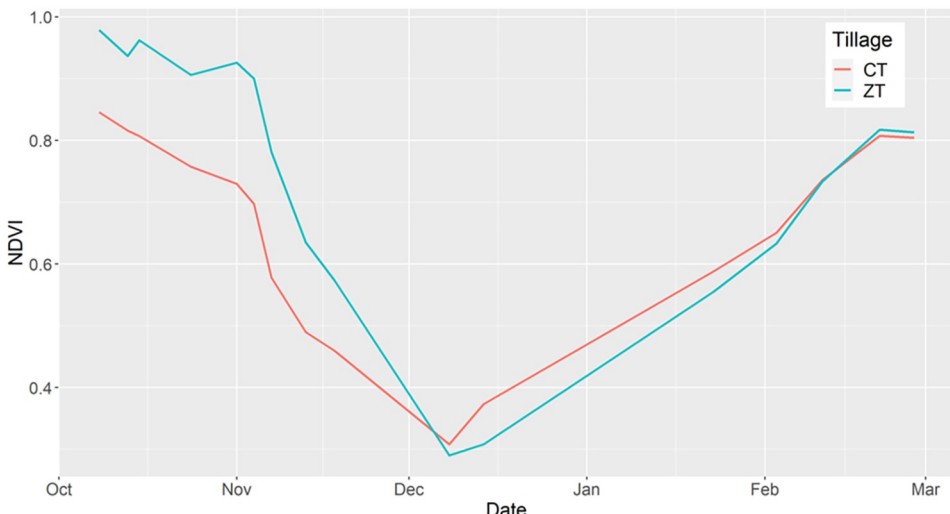

**Fig 3. Phenology curves generated from averaged NDVI values of all ZT (red) and CT (blue) fields for the 2017–18 growing season using Sentinel-2 imagery.** The drop in NDVI values from October to December represents the end of the rice crop, and captures senescence, harvest, and the removal of residues. The increase in NDVI values from December until March represents wheat biomass growth after planting.

fields. This is because ZT machinery allows farmers to plant wheat seeds into existing rice stubble, whereas in CT fields rice residue is removed by harvesting or by being incorporated into the soil through tilling.

To analyze the effectiveness of higher-resolution, readily-available satellite imagery for detecting tillage practices in smallholder systems, we used images from three different satellite sensors: 1) Synthetic Aperture Radar (SAR) Sentinel-1 [36], 2) multi-spectral Sentinel-2 [37], and 3) multi-spectral PlanetScope [38]. We obtained Sentinel-1 and Sentinel-2 data through Google Earth Engine (GEE) [39] and Planet imagery through the Planet API [40].

We obtained 13 Sentinel-1 C-band Level-1 Ground Range Detected images during our study period (Table 1). They were acquired on a descending orbit in Interferometric Wide swath mode (IW). Prior to ingestion into GEE, the data were preprocessed using the Sentinel-1 Toolbox [41]. Since speckle filtering was not done prior to ingestion, we implemented speckle filtering using the Refined Lee speckle filter code developed by Guido Lemoine (https://code. earthengine.google.com/2ef38463ebaf5ae133a478f173fd0ab5) and converted backscatter values to decibels using a logarithmic transformation. The bands and indices we used from Sentinel-1 are shown in Table 2. The intensity cross-ratio (CR) VV/VH is included as previous studies have shown its utility for differentiating vegetation types [42]. We resampled all Sentinel-1 images to 3 m resolution to match fine-scale Planet data using bilinear interpolation in GEE.

We obtained 11 Sentinel-2 Level-1C Top-Of-Atmosphere (TOA) scenes from GEE for our defined study period (Table 1). We only selected images that had less than 10% cloud cover, and visually inspected all selected images to ensure that there was no cloud cover over our field polygons. We then applied surface reflectance (SR) correction to all tiles in GEE using the radiative transfer emulator Second Simulation of the Satellite Signal in the Solar Spectrum (6S) [43] and code from Sam Murphy [44] to derive the surface reflectance values for the visible (blue, green, and red at 10 m), NIR (10 m), red-edge (20 m), and SWIR (20 m) bands. The 6S algorithm generates interpolated look-up tables (LUTs) under different atmospheric conditions considering solar zenith, ozone, and surface altitude. These LUTs are then used to calculate atmospheric correction coefficients which convert TOA radiance to SR. The bands and indices that we used from Sentinel-2 are shown in Table 2. We omitted B1, B9 and B10 because these bands represent atmospheric features, including aerosols, water vapor, and cirrus, and are not measures of the surface reflectance of land features. Eight common spectral indices were included as they have been shown to help differentiate vegetation and/or tillage practices in the previous literature (Table 2). All Sentinel-2 images and bands (10 m and 20 m) were

**Table 1. List of acquisition dates of satellite images used in our study.**

| Dataset | Sentinel-1 | Sentinel-2 | Planet |
|---|---|---|---|
| **Sowing season Peak season** | 10/03/2017 10/15/2017 10/27/2017 11/08/2017 11/20/2017 12/02/2017 12/14/2017 12/26/2017 | 10/08/2017 10/23/2017 10/28/2017 11/12/2017 11/22/2017 12/02/2017 12/12/2017 | 10/08/2017 10/13/2017 10/15/2017 10/24/2017 11/01/2017 11/04/2017 11/07/2017 11/13/2017 11/18/2017 12/08/2017 12/14/2017 |
| | 01/07/2018 01/19/2018 01/31/2018 02/12/2018 02/24/2018 | 01/31/2018 02/05/2018 02/15/2018 02/20/2018 | 01/23/2018 02/03/2018 02/11/2018 02/21/2018 02/27/2018 |

**Table 2. Band and index information for the three sensors used in this study.**

| Sensor | Spectral Index & Band | Description (Mean Wavelength: μm) | Reference |
|---|---|---|---|
| Sentinel-1 | VV | vertical transmit/vertical receive | |
| | VH | vertical transmit/horizontal receive | |
| | CR | Log ratio of (VV/VH) | [42] |
| Sentinel-2 | B2 | Blue (0.490) | |
| | B3 | Green (0.560) | |
| | B4 | Red (0.665) | |
| | B5 | Red Edge 1 (0.705) | |
| | B6 | Red Edge 2 (0.740) | |
| | B7 | Red Edge 3 (0.783) | |
| | B8 | NIR (0.842) | |
| | B8A | Red Edge 4 (0.865) | |
| | B11 | SWIR 1(1.610) | |
| | B12 | SWIR 2 (2.190) | |
| | NDTI | (SWIR1 −SWIR2) / (SWIR1 + SWIR2) | [48] |
| | CRC | (SWIR1 −Green) / (SWIR1 + Green) | [49] |
| | NDVI | (NIR—Red) / (NIR + Red) | [50] |
| | GCVI | (NIR / Green) − 1 | [51] |
| | OSAVI | (NIR—Red) / (NIR + Red + 0.16) | [52] |
| | NDI5 | (NIR—SWIR1) / (NIR + SWIR1) | [48] |
| | NDI7 | (NIR–SWIR2) / (NIR + SWIR2) | [48] |
| | STI | SWIR1 / SWIR2 | [53] |
| Planet | B1 | Blue (0.485) | |
| | B2 | Green (0.545) | |
| | B3 | Red (0.630) | |
| | B4 | NIR (0.820) | |
| | GCVI | (NIR / Green) − 1 | [51] |
| | OSAVI | (NIR—Red) / (NIR + Red + 0.16) | [49] |
| | NDVI | (NIR—Red) / (NIR + Red) | [50] |

resampled to 3 m resolution to match fine-scale Planet data using bilinear interpolation in GEE. We used bilinear interpolation as it has been shown to better preserve distributions of band values compared to other common resampling approaches [45].

We obtained 16 low-cloud Level-3B surface reflectance Planet images (Table 1) that had been atmospherically corrected by Planet using the 6S radiative transfer model with ancillary data from MODIS [38]. We defined low-cloud images as those with less than 5% cloud cover, and we further visually inspected all filtered images to ensure that there was no cloud cover over our field polygons. All individual image tiles were mosaicked using color matching of the overlapping regions using the raster package [46] in R Project Software [47]. We overlaid all Planet imagery over Sentinel-2 imagery and used visual inspection to ensure the georeferencing of these two datasets matched as previous studies have found that early iterations of Planet data needed additional georeferencing [19]. We found that the images aligned well and did not need additional georeferencing. Previous studies have shown that there is still noise remaining in the Planet surface reflectance corrected data downloaded directly from Planet [19, 20]. Thus, we conducted an additional correction by histogram stretching Planet data to match Sentinel-2 imagery using methods from Jain et al. [19] (S1 Fig). The bands and indices that we used from Planet are shown in Table 2. We computed the same indices as those calculated from Sentinel-2 using the red, green, and NIR bands (Table 2).

## Methods

### Sampling strategy, feature selection, and model development

We used random forest (RF), an ensemble-based algorithm, to classify ZT versus CT fields. Previous studies have shown that random forest often leads to high classification accuracies

**Table 3. Feature components of different sensor and sensor combinations and full and sowing season models.**

| Model | Sensor & Sensor combinations | No. of Features | No. of Selected Features | Feature Components |
|---|---|---|---|---|
| Full Model | Sentinel-1 | 39 | 15 | (2 bands + 1 index) × 13 dates |
| | Sentinel-2 | 198 | 56 | (10 bands + 8 indices) × 11 dates |
| | Planet | 112 | 34 | (4 bands + 3 index) × 16 dates |
| | Sentinel-1 + Sentinel-2 | 237 | 61 | (2 bands + 1 index) × 13 dates + (10 bands + 8 indices) × 11 dates |
| | Sentinel-1 + Planet | 151 | 45 | (2 bands + 1 index) × 13 dates + (4 bands + 3 index) × 16 dates |
| | Sentinel-2 + Planet | 310 | 77 | (10 bands + 8 indices) × 11 dates+ (4 bands + 3 index) × 16 dates |
| | Sentinel-1+ Sentinel-2 + Planet | 349 | 88 | (2 bands + 1 index) × 13 dates + (10 bands + 8 indices) × 11 dates+ (4 bands + 3 index) × 16 dates |
| Sowing Model | Sentinel-1 | 24 | 11 | (2 bands + 1 index) × 8 dates |
| | Sentinel-2 | 126 | 38 | (10 bands + 8 indices) × 7 dates |
| | Planet | 77 | 25 | (4 bands + 3 index) × 11 dates |
| | Sentinel-1 + Sentinel-2 | 150 | 41 | (2 bands + 1 index) × 8 dates + (10 bands + 8 indices) × 7 dates |
| | Sentinel-1 + Planet | 101 | 30 | (2 bands + 1 index) × 8 dates + (4 bands + 3 index) × 11 dates |
| | Sentinel-2 + Planet | 203 | 49 | (10 bands + 8 indices) × 7 dates+ (4 bands + 3 index) × 11 dates |
| | Sentinel-1+ Sentinel-2 + Planet | 222 | 55 | (2 bands + 1 index) × 8 dates + (10 bands + 8 indices) × 7 dates+ (4 bands + 3 index) × 11 dates |

with less computation time compared to other active learning models, such as support vector machines [54]. For training and validation, we used 70% of the field polygons as our training data and 30% of the field polygons as our validation data. These validation polygons were completely separate from those polygons used to train, and provide an independent source of data for validation. To ensure even representation in our training dataset regardless of field size, we randomly sampled twenty pixels from each field; for fields that were smaller than twenty pixels, we considered all available pixels within that field. In addition, we sampled an equal ratio of ZT to CT fields in both our training and validation datasets. To reduce the effect of multicollinearity on our analyses given the large number of features considered in our models, we removed highly correlated features (r > 0.9) using the caret package [55] in R Project Software. We set the number of trees for RF parameters as 500 and the number of features as $\sqrt{p}$, where p equals the number of features in the dataset. All RF classifier operations were run using the package randomForest [56] in R Project Software. The input datasets for all seven models for sowing season and full season analyses are shown in Table 3.

We evaluated our model using bootstrap analysis for 400 iterations as previous work has shown this leads to results with a 95% confidence level and avoids a power loss of more than 1% [57]. Given that each bootstrap iteration produces a different set of selected features, we averaged results across all 400 models (average number of selected features shown in Table 3). To calculate variable importance, we identified the top five most important features as ranked by the average mean decrease in accuracy. Given that all 400 bootstrap models produced different results, we present the five most important variables found across all models by averaging the mean decrease in accuracy for each variable across all models, and ranking them from the greatest to smallest value.

## Impact of spatial, temporal, and spectral resolution

To better understand the effect of improved spatial, temporal, or spectral resolution on classification accuracies, we examined the individual contribution of each in our models. First, to assess the contribution of spatial resolution on classification accuracies, we resampled the spatial resolution of Planet (3 m) to 10 m to match the spatial resolution of Sentinel-2 imagery using bilinear interpolation in GEE. We reran our single sensor Planet model using the aggregated, coarser resolution (10 m) data, and compared model results with those from the model using the original Planet data (3 m). Second, to identify the impact of improved temporal resolution on classification accuracy, we reduced the number of images used in our Planet analysis to only those dates that were similar to those available with Sentinel-2 data (Table 1). We then reran our single sensor Planet model using these limited dates (7 dates), and compared the results of this model with those from the original Planet model that included all available image dates (11 dates). Finally, to assess the impact of increased spectral information on classification accuracies, we reduced the number of bands and indices used in the single sensor Sentinel-2 model to match those used in the single sensor Planet model (Table 2). We then reran our single sensor Sentinel-2 model using these limited spectral bands and indices (7 bands and indices), and compared the results of this model with those from the original Sentinel-2 model that included all available bands and indices (18 bands and indices). We focused only on the sowing period for running these comparison models.

## Results

### Identifying which sensor or sensor combination results in the highest accuracies and variable importance

Table 4 shows mean overall accuracy and 95% confidence intervals from each of our models for the full study period. Considering which sensor and sensor combinations led to the highest classification accuracy (research question 1), we found that for single sensor models, Planet led to the best performing model. The Planet model obtained accuracies that were 3–5% higher than the next best performing single sensor model that used Sentinel-2 data. Findings indicated that the model that used only Sentinel-1 data performed poorly, with accuracies at least 20% lower than models using Sentinel-2 or Planet. Combining Sentinel-2 data and Planet data led to higher accuracies than individual sensor models, though this two-sensor model had an increase in accuracy of only 1% compared to the Planet model. Adding Sentinel-1 data did little to improve classification accuracy, and in many cases reduced overall accuracy compared to individual sensor models that used Sentinel-2 or Planet imagery, similar to Azzari et al. [14]. The highest classification accuracy in both the sowing period and full period models was obtained with the three-sensor model. The accuracies of the three sensor models, however,

**Table 4. Classification results and 95% confidence intervals for single and multiple sensor models for the full study period (Oct—Mar).**

| Sensor & Sensor combinations | Full Model Overall Accuracy | Width of 95% Confidence Interval |
|---|---|---|
| Sentinel-1 | 62.28% | 1.69% |
| Sentinel-2 | 83.24% | 2.37% |
| Planet | 86.55% | 1.77% |
| Sentinel-1 + Sentinel-2 | 82.65% | 1.86% |
| Sentinel-1 + Planet | 86.39% | 1.81% |
| Sentinel-2 + Planet | 87.61% | 1.85% |
| Sentinel-1 + Sentinel-2 + Planet | 87.71% | 1.88% |

**Table 5. Top five importance features for single and multiple sensor models for the full study period (Oct—Mar).**

|   | S1 | S2 | PS | S1+PS | S2+PS | S1+S2 | PS+S1+S2 |
|---|---|---|---|---|---|---|---|
| 1 | 1120_VH | 0131_B2 | 1008_B1 | PS_1008_B1 | PS_1008_B1 | S2_0131_B2 | PS_1008_B1 |
| 2 | 0131_VH | 1008_B2 | 1015_B1 | PS_1008_GCVI | PS_1015_B1 | S2_1112_B5 | PS_1015_B1 |
| 3 | 1202_VH | 1023_GCVI | 1008_GCVI | PS_1015_B1 | PS_1008_GCVI | S2_1023_GCVI | PS_1008_GCVI |
| 4 | 0131_VV | 1112_B5 | 1104_B4 | PS_1015_GCVI | S2_0131_B1 | S2_1008_B2 | S2_0131_B2 |
| 5 | 1120_CR | 0131_CRC | 1113_B1 | PS_1113_B1 | S2_1008_B2 | S2_0131_CRC | PS_1015_GCVI |

were only 1% better than the single sensor model that used Planet data. It is important to note that when considering 95% confidence intervals, many of the differences between the best performing single sensor and multiple sensor models were not significant. Most importantly, the Planet single sensor model performed similarly to most of the two sensor models and the three sensor model. The Planet single sensor model, however, did outperform the single sensor Sentinel-1 and Sentinel-2 models, especially for the sowing season models.

Table 5 shows the features that ranked in the top five most important of all features considered for each sensor and sensor combination for the full study period. For the models using PlanetScope data, band 1 (blue) from October 8th was always the most important feature, followed by band 1 from October 15th and GCVI from October 15th. Considering models built using Sentinel-2 data, important features ranged throughout the growing season, from early October to the end of January. Band 2 (blue) appeared to be the most common band ranked in the top five across all models that included Sentinel-2. Regarding the Sentinel-1 models, images similarly spanned the length of the growing season, from mid-November until late January. When incorporating multiple sensor data into classification models, Planet bands were largely selected as the top most important variables when the sensor was included in multi-sensor models. Bands from Sentinel-2 were the second most frequently selected important variables, though they were always less important than Planet variables when both sensors were included. Bands from Sentinel-1 never appeared in the top most important variables in multi-sensor models.

## The accuracy of using only early season imagery

Considering whether using only images from the sowing period could lead to high classification accuracies (research question 2), we found that models that used only image dates from the sowing period obtained accuracies that were very similar to the full season model, usually within a 1% difference that was smaller than the 95% confidence intervals (Table 6). This was especially true for the single or multiple sensor models that included Planet data. The biggest differences between the sowing versus full period models were seen for models that used

**Table 6. Classification results and 95% confidence intervals for single and multiple sensor models for the sowing study period (Oct—Dec).**

| Sensor & Sensor combinations | Sowing Model Overall Accuracy | Width of 95% Confidence Interval |
|---|---|---|
| Sentinel-1 | 61.24% | 1.79% |
| Sentinel-2 | 80.8% | 2.42% |
| Planet | 85.78% | 1.76% |
| Sentinel-1 + Sentinel-2 | 79.95% | 1.99% |
| Sentinel-1 + Planet | 86.03% | 1.88% |
| Sentinel-2 + Planet | 86.93% | 1.93% |
| Sentinel-1 + Sentinel-2 + Planet | 86.84% | 1.90% |

**Table 7. Classification results and 95% confidence intervals for low and high spatial, temporal, and spectral resolution models.**

| Changing Resolution | Low Resolution Model Overall Accuracy | Width of 95% Confidence Interval | High Resolution Model Overall Accuracy | Width of 95% Confidence Interval |
|---|---|---|---|---|
| Spatial | 81.32% (Planet images aggregated to 10 m) | 2.33% | 85.78% (Planet images at 3 m) | 1.76% |
| Temporal | 84.55% (7 Planet scenes) | 1.84% | 85.78% (11 Planet scenes) | 1.76% |
| Spectral | 80.17% (Sentinel-2 images with 7 bands and indices) | 2.33% | 80.80% (Sentinel-2 images with 18 bands and indices) | 2.42% |

Sentinel-2 data, with overall accuracies decreasing by approximately 3% for the sowing date model compared to the full model. These results suggest that using only images during the sowing season is as effective as using images throughout the growing season for mapping tillage practices in this region.

## The impact of improved spatial, temporal, and spectral resolution on classification accuracies

Finally, considering the impact of improved spatial, temporal, and spectral resolution (research question 3), we found that the model that used Planet satellite data aggregated to 10 m resolution led to a reduction in accuracy of 4.5% compared to the original Planet model using 3 m resolution data (Table 7). This result suggests that improved spatial resolution (3 m vs 10 m) moderately increases the accuracy of mapping field-level tillage practices. With regard to temporal resolution, we found that the model that used 7 Planet scenes had a reduction in accuracy of approximately 1% compared to the Planet model that used all 11 available scenes (Table 7). This result shows that increased temporal resolution from Planet does little to improve classification accuracy. Finally, with respect to spectral resolution, we found that the model that used Sentinel-2 data with only the bands and indices available with Planet led to a reduction in accuracy of 0.5% compared to the original Sentinel-2 model (Table 7). This result suggests that the increased spectral resolution of Sentinel-2 does not play a significant role in mapping field-level tillage practices. Considering 95% confidence intervals, only the difference due to a change in spatial resolution (10 m vs 3 m) was significant.

## Discussion

Our study examined which satellite sensor and sensor combinations as well as time periods resulted in the highest classification accuracies for mapping tillage practices for smallholder farms in the Eastern Indo-Gangetic Plains in India. We found that models that included Planet data led to the highest classification accuracies, and that models that included Sentinel-1 led to the lowest classification accuracies. Though previous studies have found that using multiple sensors can lead to more accurate classification of tillage practices [14, 58], our results showed that the best performing two sensor and three sensor models only improved accuracies by approximately 1% compared to models that only used Planet data, and this difference was not significant considering 95% confidence intervals. This suggests that in the case of smallholder farms, Planet data alone may be able to effectively map tillage practices, at least during dry growing seasons with limited cloud cover as found in our study. Considering time periods, models built using data from the sowing period were as effective as models that used data throughout the growing season. This suggests that it may be possible to map tillage practices with high accuracy after sowing has ended, providing the ability to produce real-time, within season maps of zero tillage practices at scale. Our results broadly show that tillage practices can

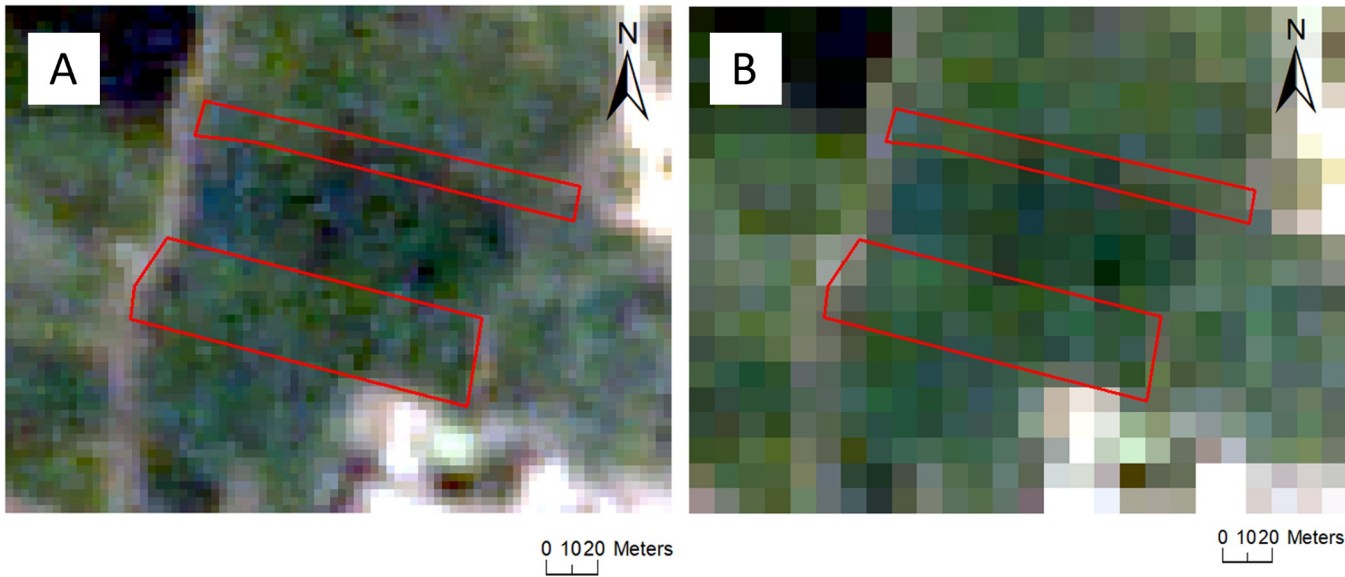

**Fig 4.** The boundaries of two field polygons overlaid on a (A) Planet image (3 m) and (B) Sentinel-2 image (10 m). Sentinel imagery was freely downloaded from the Copernicus Open Access Hub (https://scihub.copernicus.eu/).

be mapped with high accuracy (> 86%) when using relatively new high-resolution, readily available satellite imagery, even in heterogeneous, smallholder systems.

We believe that the main reason Planet imagery performed better than imagery from other sensors is due to its higher spatial resolution. This is because our analyses that examined the individual effect of improved spatial, temporal, and spectral resolutions found that reduced spatial resolution led to the greatest change in model accuracies (4.5% compared to ~ 1%). The reason improved spatial resolution is likely important for model accuracy is because Planet's improved spatial resolution of 3 m leads to fewer mixed pixels than when using coarser Sentinel-2 imagery (10 m resolution; Fig 4; S2 Fig) given the small size of fields within our study region (< 0.3 ha). Interestingly, even though Planet also has improved temporal coverage compared to Sentinel-2, this increased temporal availability did not largely increase accuracy (~ 1%). Sentinel-2 data led to models with moderate accuracy, and these models were ~5% lower in accuracy compared to Planet models. Overall we found that Sentinel-1 led to low classification accuracies and did little to improve multi-sensor model accuracies. Although Sentinel-1 provides complementary information, such as surface moisture and roughness, to optical data, optical data are much better able to discriminate differences between zero and conventional tillage. Therefore, models that include Sentinel-1 imagery probably led to reduced accuracy compared to optical-only models (Table 4) because less helpful radar data was selected at some tree nodes in these models. These results are similar to those found by Azzari et al. [14], who mapped tillage practices across the United States Midwest.

We also found that models that relied on data from the sowing season had similar accuracies to models that used data from the full study period. The importance of early season imagery was also evidenced in variable importance estimates for the full growing season (Table 5). Particularly for models that used Planet, early season imagery from October and early November were the most important variables. This suggests that the factors that are most important for distinguishing between ZT and CT likely occur during the field-preparation and sowing periods. Mechanistically this makes sense given that ZT fields are often covered in crop residue in this region, while CT fields are bare. This is because under ZT farmers do not till the soil

and can plant wheat seeds within the remaining rice residue from the previous season. This residue may lead to higher NDVI values in ZT fields compared to CT fields (Fig 3) due to remaining green vegetated biomass from the prior rice harvest [59]. Furthermore, we found that the blue bands from Planet and Sentinel-2 were often the most important predictors, likely because data from the blue wavelength have been shown to effectively detect soil properties [60] and distinguish between soil and vegetation cover [61].

When interpreting findings, there are several considerations that should be noted. First, we predicted only a binary variable of ZT versus CT, instead of a continuous variable representing tillage intensity. In reality, farmers who practice CT have heterogeneous management, with farmers varying the number of times they till their fields. Previous studies have found that it is possible to accurately classify tillage intensity of large-scale farms [14], and future work should explore whether this is also possible in smallholder systems. Second, we conducted our study during the largely dry winter growing season which has limited cloud cover compared to India's main growing season during the monsoon. It is possible that which sensor(s) lead to the highest classification accuracies may differ during cloudy seasons where optical image availability is more limited. Previous studies, for example, have shown that Sentinel-1 becomes more important for improving classification accuracies during periods of high cloud cover when optical imagery are unavailable [62]. This is largely because studies have shown that Sentinel-1 C-band data can appropriately detect vegetation phenologies across a wide range of land-cover types [63, 64], and our data suggest that ZT versus CT fields have distinct vegetation phenologies (Fig 3), particularly during the early part of the growing season. Third, our conclusions are only based on the results of one classification model, random forest, and future studies should assess whether other classification models can lead to improved accuracies. Finally, our study is limited in spatial and temporal scale; we only applied our analysis to one cropping system (rice-wheat), in one year (2017–18), and in one region (Arrah district, Bihar). Future work should examine how generalizable our findings are to other rice-wheat cropping systems in India and across multiple years. A recent study has shown that Sentinel-2 can be used to accurately map tillage practices in rice-wheat systems across Northern India over multiple years [65]. More broadly, future work should assess how generalizable our findings are to different smallholder farming systems with different cropping patterns in other parts of the world.

## Conclusions

In this study, we assessed the ability of three readily-available, high-resolution sensors (Planet, Sentinel-2, and Sentinel-1) to detect zero tillage and conventional tillage across smallholder farming systems in the Eastern Indo-Gangetic Plains in India. We find that it is possible to use readily-available, high spatial resolution satellite data to map tillage practices of smallholder farms. In particular, Planet satellite data resulted in high classification accuracy models (> 86%) and including data from additional sensors did little to improve accuracies. Tillage practices can also be mapped effectively using only data from the period of sowing, suggesting that real-time, within season maps of tillage can be produced at scale. Our work highlights the important role of micro-satellite data to map agricultural characteristics of smallholder farms, which is exciting given that the temporal resolution of such imagery is only expected to increase over the coming years as additional satellites are launched.

## Supporting information

**S1 Fig.** Histograms for Sentinel-2 (shown as blue), raw PlanetScope (shown as red), and histogram-matched PlanetScope (shown as grey) for images from October 8, 2017 for the (a) Blue,

(b) Green, (c) Red, and (d) NIR bands.
(TIF)

**S2 Fig. RGB images of Sentinel-2 and PlanetScope with some ZT/CT fields overlaid on imagery from October, 8, 2017 and VH band image of Sentinel-1 on November, 8, 2017.**
(TIF)

**S1 File. Survey conducted with farmers in the study.**
(DOCX)

## Acknowledgments

We thank the CSISA-CIMMYT field team who collected the field polygon and survey data in Bihar.

## Author Contributions

**Conceptualization:** Balwinder Singh, Amit K. Srivastava, Shishpal P. Poonia, Meha Jain.

**Data curation:** Amit K. Srivastava, Shishpal P. Poonia.

**Formal analysis:** Yin Liu.

**Funding acquisition:** Meha Jain.

**Methodology:** Yin Liu, Preeti Rao, Weiqi Zhou, Meha Jain.

**Writing – original draft:** Yin Liu.

**Writing – review & editing:** Preeti Rao, Weiqi Zhou, Balwinder Singh, Amit K. Srivastava, Shishpal P. Poonia, Derek Van Berkel, Meha Jain.

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
