## [Decision Letter · Decision Letter 0]

31 May 2022

PONE-D-22-11912Using Sentinel-1, Sentinel-2, and Planet Satellite Data to Map Field-Level Tillage Practices in Smallholder SystemsPLOS ONE

Dear Dr. Jain,

Thank you for submitting your manuscript to PLOS ONE. After careful consideration, we feel that it has merit but does not fully meet PLOS ONE’s publication criteria as it currently stands. Therefore, we invite your attention to the queries raised by the reviewers and to submit a revised version of the manuscript that addresses the points raised during the review process.

We look forward to receiving your revised manuscript.

Kind regards,

Jaishanker Raghunathan Nair, Ph.D.

Academic Editor

PLOS ONE

Journal Requirements:

a. You may seek permission from the original copyright holder of Figure(s) [#] to publish the content specifically under the CC BY 4.0 license.  

6. Please upload a new copy of Figure 4 as the detail is not clear. Please follow the link for more information: https://blogs.plos.org/plos/2019/06/looking-good-tips-for-creating-your-plos-figures-graphics/" https://blogs.plos.org/plos/2019/06/looking-good-tips-for-creating-your-plos-figures-graphics/

Additional Editor Comments:

A fairly well presented work. Please ensure the technical queries of the reviewers are addressed before final acceptance.

Reviewers' comments:

Reviewer's Responses to Questions

**Comments to the Author**

1. Is the manuscript technically sound, and do the data support the conclusions?

Reviewer #1: Yes

Reviewer #2: Yes

2. Has the statistical analysis been performed appropriately and rigorously? 

Reviewer #1: Yes

Reviewer #2: Yes

3. Have the authors made all data underlying the findings in their manuscript fully available?

Reviewer #1: Yes

Reviewer #2: Yes

4. Is the manuscript presented in an intelligible fashion and written in standard English?

Reviewer #1: Yes

Reviewer #2: Yes

5. Review Comments to the Author

Reviewer #1: Introduction:

There is relevance and clarity in what authors wish to investigate.

Materials and Methods:

The appropriateness of duration of Ground truth data, which were collected during April to May 2018 (after harvesting of the season being investigated), for winter crop may be justified (realising fully well that it is one of the most difficult information to get).

There is clarity and correctness in the method adopted.

Results: Line 304 (“... in many cases reduced overall accuracy compared to individual sensor models that...”) – Plausible reasons may be forwarded. Is it because field data is not adequate (enough)? If one were to have more field data, would such things happen (a scenario similar to Hughe’s effect starting to become evident)?

Similar comment for Table 4 wherein Sentinel - 2 accuracy is greater than Sentinel 1 + Sentinel 2 (The reason for this decrease in accuracy may be provided / speculated.).

Line 320: What causes blue band of PlanetScope to come out as the most valuable for discrimination (“...PlanetScope data, band 1 (blue) from October 8th was always the most important feature,...”) needs to be discussed. Could it be due to (relatively) poor atmospheric correction of blue band (lowest wavelength)?

Discussion and conclusion:

The results are logical, corrigible and supported by a comprehensive analysis.

Reviewer #2: This is a very nice paper assessing the usefulness of Planet, Sentinel-2 and Sentinel-1 satellite images for classifying zero tillage (ZT) vs conventional tillage (CT) field management practices in a region in the Indo-Gangetic Plains in India. The authors have done a good job in collecting a large number of satellite images and in situ data, which allows to draw solid conclusions. My only major comment is that the authors should stress the limitations of the study even more than already done. In the end, the study just considers wheat fields in the winter season 2017-18 in this region. The amount of crop residue left from the monsoon season seems to be the main indicator based on which it is possible to distinguish ZT and CT. This might be different in other regions, seasons or crops.

Specific comments

The paper is overall very well written. Yet sometime it is used the same phrases in a repetitive manner: e.g. “We used …” three times in the lines 239-243, or “results suggests …” also three times from lines 355 to 363. But there are many more examples. So please go through the text and try to reduce these repetitions.

Line 164: “Full range of” what?

Figure 3: Explain in the accompanying text already here why NDVI is higher for ZT than CT in October and November.

Line 272: Delete “conducted analyses that”

Line 402: “Who” instead of “which”

Line 237: It is problematic to state “We believe that other ML algorithms would produce other results”. Personally, I think you are right but you cannot know until you do it.

6. PLOS authors have the option to publish the peer review history of their article (what does this mean?). If published, this will include your full peer review and any attached files.

Reviewer #1: **Yes: **Markand Oza

Reviewer #2: No

---

## [Author Response · Author response to Decision Letter 0]

6 Oct 2022

Editor

Thank you for this point. We have ensured that our manuscript’s style matches the PLOS ONE style template. 

To address several of the reviewers’ points, we have added 3 references to our manuscript 

60. Lopez-Granados, F.; Jurado-Exposito, M.; Pena-Barragan, J.M.; Garcia-Torres, L. Using geostatistical and remote sensing approaches for mapping soil properties. European Journal of Agronomy 2005, 23, 3, 279-289.

61. USGS. Mapping, Remote sensing, and Geospatial data. https://www.usgs.gov/faqs/what-are-best-landsat-spectral-bands-use-my-research. Accessed June 21, 2022.

65. Zhou, W.; Rao, P.; Jat, M.L.; Singh, B.; Singh, R.; Schulthess, U.; Poonia, S.; Bijarniya, D.; Singh, L.K.; Kumar, M.; Jain, M. Using Sentinel-2 to Track Field-Level Tillage Practices at Regional Scales in Smallholder Systems. Remote Sensing 2021, 13, 24, 5108

Thanks for this important point. We have now included all minimal data sets in the Zenodo data repository (https://doi.org/10.5281/zenodo.6703973). We have added this to the data availability statement at the end of our manuscript. 

Thanks, we have now added the ORCID iD for the corresponding author, Meha Jain (0000-0002-6821-473X).

 a. You may seek permission from the original copyright holder of Figure(s) [#] to publish the content specifically under the CC BY 4.0 license. 

Thank you. We have removed the basemap that was potentially of copyright worry, and now should have no issues. The image shown in panel C is high-res imagery used for our analysis obtained from Planet data and we made the image ourselves. 

6. Please upload a new copy of Figure 4 as the detail is not clear. Please follow the link for more information: https://blogs.plos.org/plos/2019/06/looking-good-tips-for-creating-your-plos-figures-graphics/" https://blogs.plos.org/plos/2019/06/looking-good-tips-for-creating-your-plos-figures-graphics/

Thanks, we have now uploaded a new Figure 4. Please let us know if this is not sufficient. 

Additional Editor Comments:

A fairly well presented work. Please ensure the technical queries of the reviewers are addressed before final acceptance.

Thank you for the chance to address your and reviewers’ comments. We believe the manuscript has been greatly improved thanks to these suggestions.

 

Reviewer #1 

Introduction:

There is relevance and clarity in what authors wish to investigate.

Thank you very much for your comments.

Materials and Methods:

The appropriateness of duration of Ground truth data, which were collected during April to May 2018 (after harvesting of the season being investigated), for winter crop may be justified (realising fully well that it is one of the most difficult information to get).

There is clarity and correctness in the method adopted.

Thank you for these comments. 

Results: 

Line 304 (“... in many cases reduced overall accuracy compared to individual sensor models that...”) – Plausible reasons may be forwarded. Is it because field data is not adequate (enough)? If one were to have more field data, would such things happen (a scenario similar to Hughe’s effect starting to become evident)?

Thank you for this interesting and important point. We believe that the reduced accuracy from Sentinel-1 is because radar data are not able to detect differences between zero tillage and conventional tillage as well as optical data. These findings are similar to those found in other studies that have used Sentinel-1 data along with optical imagery to map tillage practices (e.g., Azzari et al. 2021). We have now stated that our results are similar to previous studies in the results section (after line 304 above), and we have discussed the reasons for this further in the discussion section. 

“Adding Sentinel-1 data did little to improve classification accuracy, and in many cases reduced overall accuracy compared to individual sensor models that used Sentinel-2 or Planet imagery, similar to Azzari et al. [4].” (page 9, lines 320-322).

“Although Sentinel-1 provides complementary information, such as surface moisture and roughness, to optical data, optical data are much better able to discriminate differences between zero and conventional tillage. Therefore, models that include Sentinel-1 imagery probably lead to reduced accuracy compared to optical-only models (Table 4) because less helpful radar data is selected at some tree nodes in these models.” (page 12, lines 427-432).

Similar comment for Table 4 wherein Sentinel - 2 accuracy is greater than Sentinel 1 + Sentinel 2 (The reason for this decrease in accuracy may be provided / speculated.).

Thanks again for this helpful comment, and we have now cited Table 4 in our explanation for why models that include Sentinel-2 have reduced accuracy. 

“Although Sentinel-1 provides complementary information, such as surface moisture and roughness, to optical data, optical data are much better able to discriminate differences between zero and conventional tillage. Therefore, models that include Sentinel-1 imagery probably lead to reduced accuracy compared to optical-only models (Table 4) because less helpful radar data is selected at some tree nodes in these models.” (page 12, lines 427-432).

Line 320: What causes blue band of PlanetScope to come out as the most valuable for discrimination (“...PlanetScope data, band 1 (blue) from October 8th was always the most important feature,...”) needs to be discussed. Could it be due to (relatively) poor atmospheric correction of blue band (lowest wavelength)?

Thanks for your interesting point. We found that the blue bands from PlanetScope and Sentinel-2 often come out to be the most important variables for detecting zero tillage versus conventional tillage. We believe this is because the blue band has been shown to distinguish between soil and vegetation cover, and has also been shown to effectively map soil properties, such as soil organic carbon, that may differ between conventional and zero tillage fields. We have added these explanations in the discussion section. 

“This suggests that the factors that are most important for distinguishing between ZT and CT likely occur during the field-preparation and sowing periods. Mechanistically this makes sense given that ZT fields are often covered in crop residue in this region, while CT fields are bare. This is because under ZT farmers do not till the soil and can plant wheat seeds within the remaining rice residue from the previous season. This residue may lead to higher NDVI values in ZT fields compared to CT fields (Fig 3) due to remaining green vegetated biomass from the prior rice harvest [59]. Furthermore, we found that the blue bands from Planet and Sentinel-2 were often the most important predictors, likely because data from the blue wavelength have been shown to effectively detect soil properties [60] and distinguish between soil and vegetation cover [61].” (page 12, lines 443-453).

Discussion and conclusion:

The results are logical, corrigible and supported by a comprehensive analysis.

Thank you for these comments. 

 

Reviewer #2 

This is a very nice paper assessing the usefulness of Planet, Sentinel-2 and Sentinel-1 satellite images for classifying zero tillage (ZT) vs conventional tillage (CT) field management practices in a region in the Indo-Gangetic Plains in India. The authors have done a good job in collecting a large number of satellite images and in situ data, which allows to draw solid conclusions. 

Thank you very much for these comments. 

My only major comment is that the authors should stress the limitations of the study even more than already done. In the end, the study just considers wheat fields in the winter season 2017-18 in this region. The amount of crop residue left from the monsoon season seems to be the main indicator based on which it is possible to distinguish ZT and CT. This might be different in other regions, seasons or crops.

Thanks for this important suggestion. We agree that our study is limited in terms of crop type, study region, and time period, and it would be worthwhile to expand the discussion of the limitations of our study and its ability to be generalized to other regions. We have expanded our discussion of limitations in the discussion section. 

“Finally, our study is limited in spatial and temporal scale; we only applied our analysis to one cropping system (rice-wheat), in one year (2017-18), and in one region (Arrah district, Bihar). Future work should examine how generalizable our findings are to other rice-wheat cropping systems in India and across multiple years. A recent study has shown that Sentinel-2 can be used to accurately map tillage practices in rice-wheat systems across Northern India over multiple years [65]. More broadly, future work should assess how generalizable our findings are to different smallholder farming systems with different cropping patterns in other parts of the world.” (page 13, lines 481-489).

Specific comments

The paper is overall very well written. Yet sometimes it is used the same phrases in a repetitive manner: e.g. “We used …” three times in the lines 239-243, or “results suggests …” also three times from lines 355 to 363. But there are many more examples. So please go through the text and try to reduce these repetitions.

Thanks for catching this. We have edited the text as suggested, and also read through the manuscript to remove other instances of repetition. 

Line 164: “Full range of” what?

Thanks. We have changed this to ‘full range of phenological change…’ (page 4, line 165).

Figure 3: Explain in the accompanying text already here why NDVI is higher for ZT than CT in October and November.

We have added the following text to address this suggestion.

“From October to December, NDVI is higher in ZT fields compared to CT fields (Fig. 3), likely because farmers maintain monsoon rice crop residue on ZT fields but not CT fields. This is because ZT machinery allows farmers to plant wheat seeds into existing rice stubble, whereas in CT fields rice residue is removed by harvesting or by being incorporated into the soil through tilling.” (page 5, lines 171-176).

Line 272: Delete “conducted analyses that”

Done.

Line 402: “Who” instead of “which”

Done.

Line 237: It is problematic to state “We believe that other ML algorithms would produce other results”. Personally, I think you are right but you cannot know until you do it.

This is a good suggestion, and we have removed this statement. Instead, we have listed our use of only random forest as a limitation. 

“Third, our conclusions are only based on the results of one classification model, random forest, and future studies should assess whether other classification models can lead to improved accuracies.” (page 13, lines 479-481).

---

## [Editor Report · Decision Letter 1]

27 Oct 2022

Using Sentinel-1, Sentinel-2, and Planet Satellite Data to Map Field-Level Tillage Practices in Smallholder Systems

PONE-D-22-11912R1

Dear Dr. Jain,

We’re pleased to inform you that your manuscript has been judged scientifically suitable for publication and will be formally accepted for publication once it meets all outstanding technical requirements.

Kind regards,

Jaishanker Raghunathan Nair, Ph.D.

Academic Editor

PLOS ONE
---

## [Editor Report · Acceptance letter]

14 Nov 2022

PONE-D-22-11912R1 

Using Sentinel-1, Sentinel-2, and Planet Satellite Data to Map Field-Level Tillage Practices in Smallholder Systems 

Dear Dr. Jain:

I'm pleased to inform you that your manuscript has been deemed suitable for publication in PLOS ONE. Congratulations! Your manuscript is now with our production department. 

Kind regards, 

on behalf of

Dr. Jaishanker Raghunathan Nair 

Academic Editor

PLOS ONE